# The six steps of the complete F$_1$-ATPase rotary catalytic cycle

Meghna Sobti[1,2], Hiroshi Ueno[3], Hiroyuki Noji[3✉] & Alastair G. Stewart [1,2✉]

F$_1$F$_o$ ATP synthase interchanges phosphate transfer energy and proton motive force via a rotary catalysis mechanism. Isolated F$_1$-ATPase catalytic cores can hydrolyze ATP, passing through six intermediate conformational states to generate rotation of their central γ-subunit. Although previous structural studies have contributed greatly to understanding rotary catalysis in the F$_1$-ATPase, the structure of an important conformational state (the binding-dwell) has remained elusive. Here, we exploit temperature and time-resolved cryo-electron microscopy to determine the structure of the binding- and catalytic-dwell states of *Bacillus* PS3 F$_1$-ATPase. Each state shows three catalytic β-subunits in different conformations, establishing the complete set of six states taken up during the catalytic cycle and providing molecular details for both the ATP binding and hydrolysis strokes. We also identify a potential phosphate-release tunnel that indicates how ADP and phosphate binding are coordinated during synthesis. Overall these findings provide a structural basis for the entire F$_1$-ATPase catalytic cycle.

[1] Molecular, Structural and Computational Biology Division, The Victor Chang Cardiac Research Institute, Darlinghurst, NSW, Australia. [2] Faculty of Medicine, St Vincent's Clinical School, UNSW Sydney, Kensington, NSW, Australia. [3] Applied Chemistry, Graduate School of Engineering, The University of Tokyo, Tokyo, Japan. ✉email: hnoji@g.ecc.u-tokyo.ac.jp; a.stewart@victorchang.edu.au

F$_1$F$_o$ ATP synthase is a biological rotary motor that utilizes a rotary catalytic mechanism to couple proton translocation across a membrane with the synthesis of adenosine triphosphate (ATP) from inorganic phosphate (P$_i$) and adenosine diphosphate (ADP)[1–4]. The enzyme is comprised two rotary motors, the F$_1$-ATPase and the F$_o$ motor (Fig. 1a), which are coupled together by two stalks, a central "rotor" stalk and a peripheral "stator" stalk. The F$_o$ motor spans the membrane and converts the potential energy from the proton motive force into mechanical rotation of the central rotor. This central rotor drives conformational changes in the catalytic F$_1$-ATPase, where ATP is synthesized from ADP and P$_i$[5,6]. Studies on the isolated F$_1$-ATPase have shown that it can also function in reverse, with ATP hydrolysis inducing rotation of the rotor[3,5,7]; the study presented here focuses primarily on the F$_1$-ATPase operating under ATP hydrolysis. The F$_1$-ATPase consists of three α-, three β-, and a single γ-subunit, with the α- and β-subunits arranged in an alternating manner (αβαβαβ) to form a hexameric ring with the γ-subunit at its center (Fig. 1a). There are six nucleotide-binding sites at the interfaces between the α- and β-subunits. Three of the nucleotide-binding sites are catalytically active, whereas the other three nucleotide-binding sites are not catalytically active and just bind MgATP. The three catalytically active sites are mainly encompassed by the β-subunits and therefore are termed the β-subunit sites. Moreover, the β-subunit is the principal catalytic subunit that undergoes conformational changes and, as they change sequentially, they drive the rotation of the γ-subunit during ATP hydrolysis[5–9]. The α-subunit is very similar in sequence to the β-subunit, but merely binds MgATP and does not hydrolyze it. The threefold arrangement of the catalytic sites dictates that the γ-subunit rotates in 120° steps, with three ATPs being turned over for each full revolution of the rotor.

Single-molecule fluorescence studies on *Bacillus* PS3 F$_1$-ATPase (hereafter termed TF$_1$—for Thermophilic F$_1$-ATPase), which could directly visualize the rotation of the γ-subunit during ATP hydrolysis, initially showed a rotation step of 120°[10]. However, further studies on TF$_1$ utilizing different conditions and improved equipment were able to resolve two discrete substeps of ~80° and ~40°[11–13], dissecting the 120° step into two strokes. The rotation dwells before the 80° and 40° substeps were termed the "binding dwell" and "catalytic dwell," respectively, because these dwells were first identified as either waiting for ATP binding[11] or waiting for hydrolysis[14,15]. Subsequently, rotation assays on TF$_1$ showed that phosphate is likely released during the 40° substep[16] and experiments using fluorescent nucleotide showed that ADP is released during the 80° step[16,17]. All these findings can be summarized into a circular reaction scheme that shows the major catalytic events linked to rotary position (Fig. 1b). In a full rotation, the F$_1$-ATPase makes conformational transitions between the binding-dwell and catalytic-dwell states, with each β-subunit undergoing six conformational transitions.

Bovine mitochondrial F$_1$-ATPase (hereafter termed bMF$_1$) has been a long-standing model system to characterize the F$_1$-ATPase structurally using X-ray crystallography. Although rich in biochemical information[5,18–20], the crystal structures of bMF$_1$ principally represent the catalytic-dwell state, or related transition states, with the position of the γ-subunit determined by crystal lattice contacts with the crown region of adjacent complexes[19]. In the "ground-state" crystal structure of bMF$_1$[18], each β-subunit contained a different nucleotide composition and was either in an "open" or "closed" conformation, with the subunits named due to the nucleotide occupancy: β$_{DP}$ contained MgADP and was in a closed state, β$_{TP}$ contained MgAMP-PNP and was in a closed state, and β$_E$ was in an open state with no nucleotide bound (Supplementary Fig. 1a). The β$_{DP}$ and β$_{TP}$ subunits had

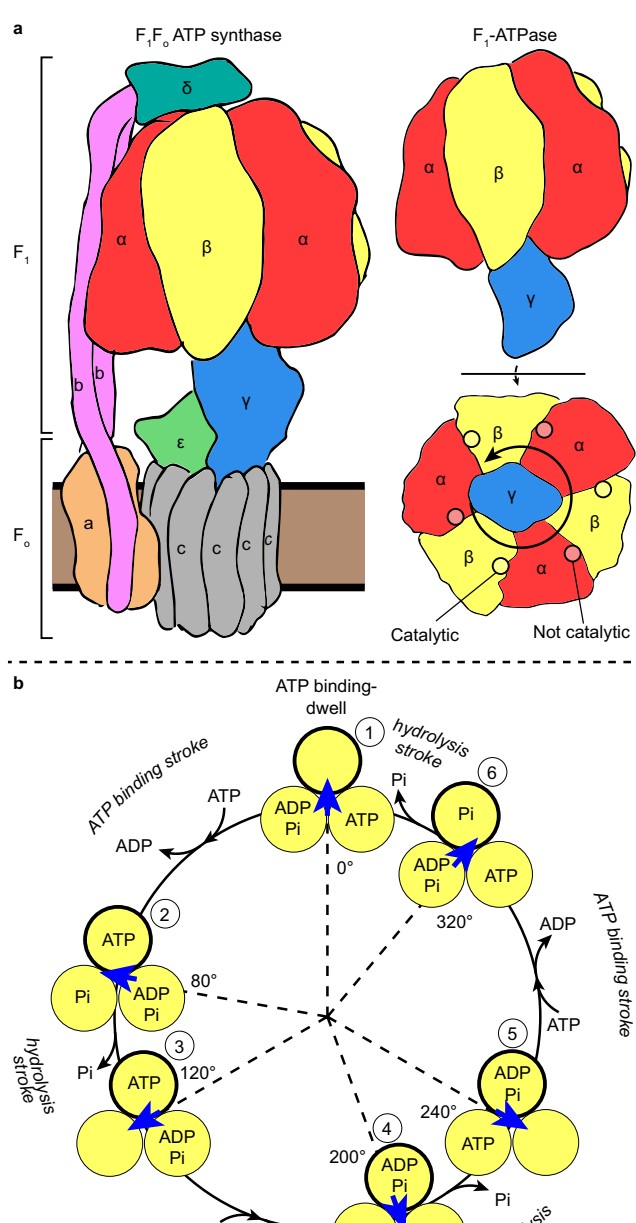

**Fig. 1 Schematic of F$_1$-ATPase architecture and function. a** Schematic to show the overall arrangement of subunits in F$_1$F$_o$ ATP synthase (left) and F$_1$-ATPase (right). Direction of γ-subunit rotation in F$_1$-ATPase during ATP hydrolysis is depicted with an arrow and the positions of the nucleotide-binding sites are highlighted and labeled (bottom right panel—viewed from the membrane, which is referred to as the "below" view hereafter). **b** Schematic of the F$_1$-ATPase rotary catalytic mechanism, as suggested by single-molecule experiments. β-Subunits are depicted as yellow circles with binding site occupancy labeled, the γ-subunit is depicted with a blue arrow, and, for clarity, the α-subunits are not shown. The F$_1$-ATPase γ-subunit rotates with six strokes alternating between 80° and 40° movements. The ATP-binding dwell precedes ATP binding and is followed by the "ATP-binding stroke," whereas the catalytic dwell precedes the "hydrolysis stroke." One of the β-subunits is outlined in black and at each dwell is numbered (from 1 to 6) to highlight the reaction path that a single subunit undergoes from the starting 0° position.

essentially the same conformation, although the αβ-interface is slightly more closed in the $\alpha_{DP}$–$\beta_{DP}$ pair. The $\beta_E$ subunit assumes the open conformation, hinging its C-terminal "foot" domain outward compared to $\beta_{DP}$ and $\beta_{TP}$. Overall, the crystal structures of $bMF_1$ in the ground state represent the catalytic-dwell state and show only three of the six conformational states of the β-subunit found in single-molecule rotation assays.

Structural studies that attempt to resolve structures other than the ground state have also been reported. In the crystal structure of $bMF_1$ solved in the presence of aluminum fluoride, the $\beta_E$ subunit binds to ADP and sulfate, adopting a "half-closed" conformation, i.e., in an intermediate conformation between the closed and open conformations[21]. The γ-subunit was rotated by 15° from the rotary position seen in the ground state[22]. Based on these features, this structure is considered to represent a pre-ADP intermediate state of the $F_1$-ATPase that appears in the 80° substep, i.e., in the conformational transition from the binding-dwell state to the catalytic-dwell state. Crystallography and cryo-electron microscopy (cryo-EM) have been used to study the autoinhibited states of $TF_1$ and *Escherichia coli* $F_1$-ATPase (hereafter termed $EF_1$), in which the β-subunits are in open, closed, open conformations in $TF_1$[23,24] and half-closed, closed, open conformations in $EF_1$[25,26]. All of these studies on $TF_1$ and $EF_1$ showed the inhibitory ε-subunit in an "up" position, which is believed to prevent rotation and thereby prevents one of the β-subunits from closing. In studies examining the isolated bacterial $F_1$-ATPase undergoing hydrolysis, such as the study presented here, the ε-subunit is usually not expressed and purified so that it does not inhibit the enzyme[7]. Crystal structures of yeast[27] and other bacterial[28] $F_1$-ATPases have all shown a similar rotational state to the $bMF_1$ ground state. To date, no structure representing the binding dwell has been obtained. Thus, the structures of three of the six conformational states of the β-subunit during ATP catalysis have not been established and it remains unclear how the β-subunit induces rotation upon ATP binding or catalysis and product release.

The higher stability of the catalytic dwell and associated states is the likely reason that all structures of $F_1$-ATPase published to date have being determined primarily in the catalytic-dwell rotational position. In particular, this can be attributable to the ADP-inhibited state, where the enzyme in known to pause catalysis and rotation at the catalytic-dwell angle[29], with the crystal structure of $bMF_1$ in this state being near-identical to the bMF1 ground-state crystal structure[30]. To address this problem, we froze and imaged $TF_1$ in a temperature-sensitive dwell state detected in single-molecule rotation assays of $TF_1$. These studies have shown that, when rotation of $TF_1$ is observed below 10 °C, the enzyme preferentially dwells at the ATP-binding angle[31]. Due to the temperature-sensitive nature of this dwell, we term this state the "binding-dwell (TS)" hereafter. A mutation in the catalytic site, βE190D in $TF_1$, has been shown to substantially extend the duration of the binding-dwell (TS)[32]. Hence, as structural studies only provide a static snapshot of a given molecule at any one time, imaging $TF_1$(βE190D) while it undergoes ATP hydrolysis at 10 °C would facilitate observation of the enzyme stalled in the binding-dwell (TS) state. Moreover, imaging the same reaction, at 28 °C, would produce a structure of the enzyme but in the catalytic dwell, which could be used as a direct comparison to understand the molecular basis of rotation in the $F_1$-ATPase.

Here we combine time[33]- and temperature[34]-resolved cryo-EM to examine the temperature-sensitive $TF_1$(βE190D) mutant under three conditions in both the catalytic-dwell and binding-dwell (TS) states of $TF_1$. In cryo-EM, it is relatively trivial to control the sample environment prior to freezing. In the present study, this is exploited by controlling the sample temperature prior to freezing to weight the populations towards either the catalytic-dwell

(incubation at 28 °C) or binding-dwell (TS) (incubation at 10 °C). Subsequently, computational methods are used to sort the particles into distinct conformational states. The cryo-EM maps indicate how ATP tightly binds to induce the 80° ATP-binding stroke and ATP hydrolysis induces the 40° hydrolysis stroke. Two intriguing features of these movements are that a fully closed β-subunit acts as a pivot for the γ-subunit to rotate around, and that the ATP hydrolysis step reorients a β-subunit to a mechanically favorable position for the subsequent ATP-binding stroke. Close inspection of the maps also identifies a secondary access tunnel that allows $P_i$ dissociation/association, even when bound ADP plugs the nucleotide-binding cleft. Taken together, these cryo-EM structures provide a molecular-level understanding on the rotary catalytic mechanism of $F_1$-ATPase.

## Results

**Structure of *Bacillus* PS3 $F_1$-ATPase in two rotational states.** Kinetic analyses from single-molecule studies estimate that at 10 °C ~80% of $TF_1$(βE190D) molecules pause at the binding-dwell angle, whereas at 28 °C ~85% pause at the catalytic angle[31,32]. Hence, to obtain cryo-EM maps of the $F_1$-ATPase pausing at both the binding-dwell and catalytic-dwell angles, MgATP was added (to a final concentration of 10 mM) to purified active $TF_1$(βE190D) (Supplementary Fig. 2), then applied to EM grids at either 10 °C or 28 °C, frozen in liquid ethane, and subsequently imaged at 300 kV followed by single-particle analysis (SPA) using standard methods (Supplementary Fig. 3a, b). There was <20 s between addition of MgATP and freezing, with 3 s to add/mix MgATP, 6 s to apply the sample to the grid, and 9 s to blot and freeze the grid. We hypothesized that, given the results of previous single-molecule observations[32], this strategy of observing $TF_1$(βE190D) after addition of MgATP and freezing the reaction from different temperatures would enable the observation of $TF_1$ in two different dwell states. SPA-sorting methods, termed three-dimensional (3D) classification or heterogenous refinement, are able to sort "picked particles" to define different 3D structures or conformations within the sample[35]. In the present study, these methods were used to separate the particles in each data set into two conformations, thereby enabling generation of cryo-EM maps of the motor in different rotational dwell states (Supplementary Fig. 3a, b). Cryo-EM maps were obtained for two different dwell states, to 3.1 and 3.4 Å resolution, which were related by a rotation of the γ-subunit (Fig. 2). Although the maps were not to atomic resolution, they provided sufficient resolution to establish the nucleotide occupancy of each binding site (Fig. 3, Supplementary Fig. 4, and Supplementary Movie 1).

The predominant cryo-EM structure of $TF_1$(βE190D) obtained when frozen from 28 °C was almost identical to the $bMF_1$ ground-state crystal structure; two of the β-subunits were in a closed conformation and bound to MgATP (akin to $\beta_{TP}$ and $\beta_{DP}$ of $bMF_1$), whereas the third β-subunit was in an open conformation (akin to $\beta_E$ or $\beta_{DP}$ of $bMF_1$) (Supplementary Fig. 1). The structure of $TF_1$ obtained when frozen from 28 °C likely corresponds to a "hydrolysis-waiting" state where the enzyme is paused just prior to hydrolysis, representing the motor in the catalytic-dwell state. Although the structure was highly similar to the crystal structure of $bMF_1$ in the ground state, one clear difference was the $\beta_E$ nucleotide occupancy. In the crystal structure of $bMF_1$ in ground state, the $\beta_E$ site was empty and did not contain any nucleotide, whereas the equivalent site in the cryo-EM structure of $TF_1$ contained MgADP.

The predominant structure obtained at 10 °C showed features distinct from the catalytic-dwell structure observed at 28 °C. Two of the β-subunits assumed conformations that are in-between the closed and open forms, whereas the third assumed a closed

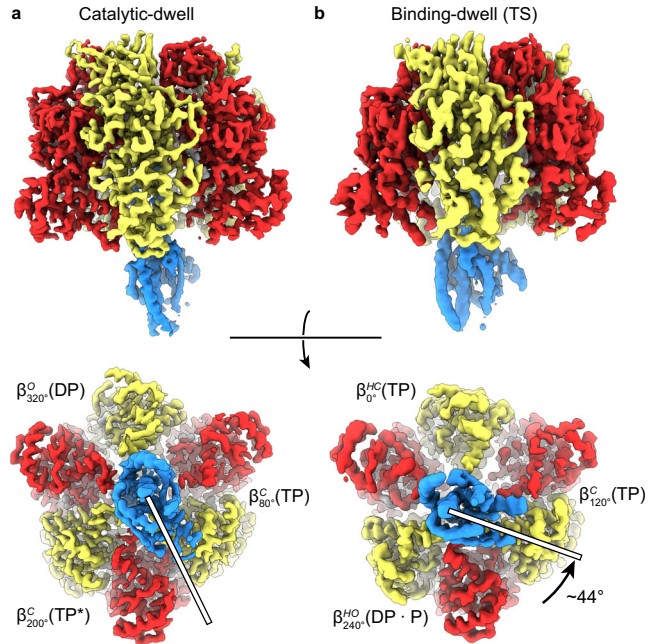

**Fig. 2 Cryo-EM maps of *Bacillus* PS3 F₁-ATPase in two rotational dwells.** Cryo-EM maps of two rotational dwells of TF₁(βE190D) viewed from the side (top) and from below (bottom), with subunits colored as in Fig. 1a. **a** The predominant structure when imaged following the addition of 10 mM MgATP at 28 °C—the catalytic dwell. **b** The predominant structure when imaged following the addition of 10 mM MgATP at 10 °C—the binding-dwell (TS). Comparison of the binding dwell with the catalytic dwell indicated that between these dwells the γ-subunit rotates ~44° in the counterclockwise direction (rotation highlighted with white bars and black arrow). Overall, each dwell provided three different conformations of the F₁-ATPase β-subunits and so provided a spectrum of the six sub-states through which the enzyme passes during its hydrolysis cycle, here termed $\beta_{0°}^{HC}$(TP), $\beta_{80°}^{C}$(TP), $\beta_{120°}^{C}$(TP), $\beta_{200°}^{C}$(TP*), $\beta_{240°}^{HO}$(DP•P), and $\beta_{320°}^{O}$(DP).

conformation (Figs. 3 and 4). We term these β-subunit conformations "half-open" and "half-closed" as the subunit would be transitioning to an open or closed state, respectively, when operating under ATP hydrolysis. These states are not equivalent conformations, because the C-terminal "foot" is not in the same relative position (Fig. 4). Another striking feature is evident when the predominant structure observed when frozen from 10 °C is compared to that frozen from 28 °C (the catalytic dwell), with the γ-subunit rotating relative to the α- and β-subunits. Fig. 2 shows the two structures determined in this study aligned on the α₃β₃ stator ring, showing a 44° rotation of the γ-subunit (as calculated with CCP4mg[36]). This agrees well with the 40° rotation observed between the catalytic dwell and binding-dwell (TS) in single-molecule studies[15,32], suggesting that the predominant conformation observed when frozen from 10 °C represents the state pausing at the binding-dwell angle, as expected. All three β-subunits had nucleotide bound, with these being ATP, ATP, and ADP + Pᵢ, with the Coulomb density corresponding to ADP + Pᵢ being elongated compared to the other ATP molecules seen in the other sites (Fig. 3, Supplementary Fig. 4, and Supplementary Movie 1). These observations are also consistent with the expected site occupancy of the binding-dwell (TS) suggested by single-molecule experiments (Fig. 1b): the β-subunit at 0° represents the state after ATP binding, but before the ATP binding 80° stroke, the β-subunit at 120° has ATP tightly bound and the β-subunit at 240° represents the post-hydrolysis state, with ADP + Pᵢ bound. Therefore, we conclude that the predominant structure observed when frozen from 10 °C

represents the binding-dwell (TS), pausing at the binding-dwell angle. The historic notation to describe the conformation/nucleotide occupancy of the β-subunits was based on just three states ($\beta_E$, $\beta_{DP}$, and $\beta_{TP}$), the results presented here enable this notation to be expanded to include all six states defining the β-subunits based on their conformation, the approximate rotational position, and the nucleotide occupancy.

$$\beta_{r°}^{c}(n)$$

With $c$ representing the conformational state of the β-subunit, $r$ representing the rotation angle from 0° (as defined in Fig. 1b), and $n$ representing the nucleotide occupancy of the β-subunit.

Hence, the β-subunits found in the binding-dwell (TS) structure are hereafter referred to as $\beta_{0°}^{HC}$(TP), $\beta_{120°}^{C}$(TP), and $\beta_{240°}^{HO}$(DP•P), respectively. Correspondingly, $\beta_{TP}$, $\beta_{DP}$, and $\beta_E$ in the catalytic dwell are termed $\beta_{80°}^{C}$(TP), $\beta_{200°}^{C}$(TP*), and $\beta_{320°}^{O}$(DP) below for consistency. With nucleotide abbreviations of TP for ATP, TP* for ATP in catalysis, DP•P for ADP and phosphate, DP for ADP, and conformation abbreviations of HC for half-closed, C for closed, HO for half-open, and O for open.

Neither data set contained one conformation exclusively, with the 28 °C cryo-EM image data set containing a ~3 : 1 and the 10 °C cryo-EM image data set containing a ~2 : 3 ratio of the catalytic- and binding-dwell (TS), respectively, as classified by cryoSPARC heterogeneous refinement[37]. Although this ratio of particles does not agree fully with single-molecule studies, the observed populational shift between the two states well reflects the temperature dependency of TF₁(βE190D), suggesting that the structural analysis was valid. The discrepancy between the relative proportions of the catalytic-dwell and binding-dwell (TS) states seen in this study vs. those in single-molecule rotation studies could be due to many factors. For example: (i) the classification of particles may be incomplete, with some particles assigned to the incorrect class or "junk" particles not corresponding to TF₁ molecules still being present. Much care was taken to classify these data to completion, but as particles are weighted they may still be present and not contribute to the final reconstruction. (ii) There could be cooling effects from rapid evaporation during the blotting process that would change the local temperature prior to freezing. (iii) As the protein molecules are likely to be in contact with the air-water interface, this external influence may affect the turnover of the enzyme. Still, having the two data sets increased the detail obtained on each dwell state, as the resolution was highest for the predominant structure at each temperature, and it allowed us to assign the structure observed at each temperature with that observed in single-molecule rotation assays.

The set of sequential conformational states observed in this study are broadly consistent with the scheme suggested by single-molecule experiments that is outlined in Fig. 1b. However, an exception is that ADP is observed bound to the $\beta_{320°}$ site, whereas previous studies, based on information from the ground-state bMF₁ crystal structure, hypothesized that this site would be empty and not contain any nucleotide. One potential interpretation of this point is that this discrepancy represents a divergence of the F₁-ATPase reaction scheme between species and the $\beta_{320°}$ conformational state represents a catalytic intermediate that has yet to be seen in single-molecule studies, an "ADP-releasing state" that is only present in TF₁. Single-molecule studies on TF₁ have shown that the β-subunit releases ADP during the transition from $\beta_{240°}$ to $\beta_{320°}$[16,17,38] but, as the precise timing could not be resolved, ADP release may occur at $\beta_{320°}$. An obvious alternative explanation is that the ADP observed in the $\beta_{320°}$ represents ADP re-binding from solution, with ADP being generated by the enzyme up until it is frozen. However, this point remains to be addressed in future work.

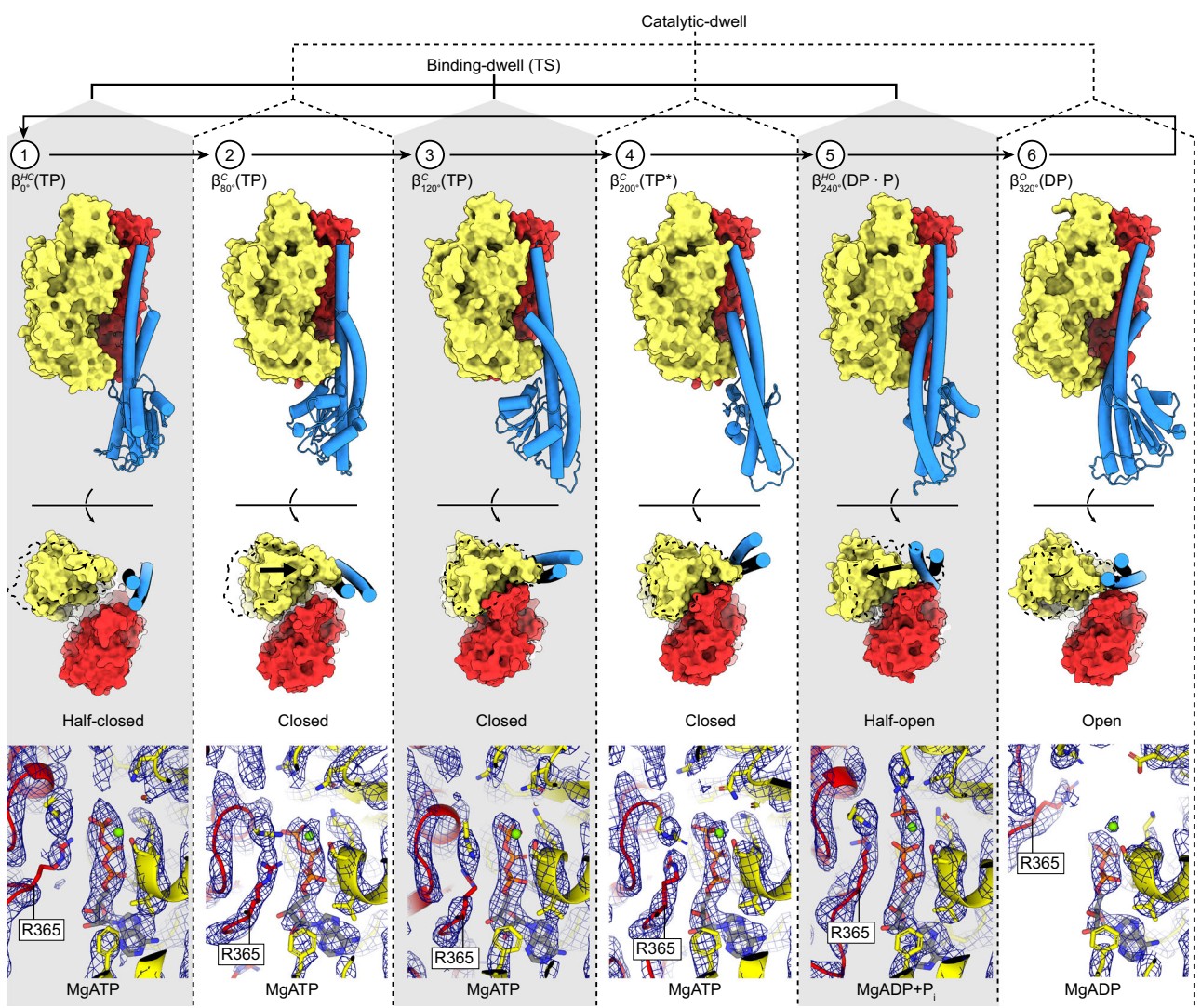

**Fig. 3 The six sequential conformations in the F$_1$-ATPase rotary catalytic cycle.** Top, αβ-pairs superposed on the N termini (β2-82) and viewed from the side (perpendicular to the membrane) and below (from the membrane). Subunit α in red, β in yellow, and γ in blue, with stencil outline of the β-subunit from the previous step in the scheme for comparison. Bottom, close-up of the catalytic nucleotide-binding sites, superimposed on residues around the nucleotides (β158-166, β336-342, and β412-421). Cryo-EM map shown as blue mesh. Nucleotides, Mg$^{2+}$, and P$_i$ are shown as sticks with CPK coloring and αR365 (the arginine finger) labeled. In the movement from the binding-dwell (TS) (states 1, 3, and 5) to the catalytic dwell (states 2, 4, and 6): 1 → 2 transition ($\beta_{0°}^{HC}$(TP) → $\beta_{80°}^{C}$(TP)), the αβ-subunits close to bind MgATP tightly. 2 → 3 → 4 Transition ($\beta_{80°}^{C}$(TP) → $\beta_{120°}^{C}$(TP) → $\beta_{200°}^{C}$(TP*)), the β-subunit remains in a similar position, with a minor movement of the α-subunit about the nucleotide. 4 → 5 Transition ($\beta_{200°}^{C}$(TP*) → $\beta_{240°}^{HO}$(DP•P)), MgATP is hydrolyzed to MgADP + P$_i$ (elongated density in panel 5), and the β-subunit opens to a half-open state, with the arginine finger (αR365) moving relative to the adenosine ring to stabilize the MgADP + P$_i$. 5 → 6 Transition ($\beta_{240°}^{HO}$(DP•P) → $\beta_{320°}^{O}$(DP)), P$_i$ is released and the β-subunit opens. 6 → 1 Transition ($\beta_{320°}^{O}$(DP) → $\beta_{0°}^{HC}$(TP)), MgADP is released, followed by MgATP binding and the β-subunit half closing to start the cycle again.

Several lines of evidence indicate that P$_i$ is released at the $\beta_{320°}$ position[16,39,40]. As the F$_1$-ATPase is known to release P$_i$ within milliseconds after catalysis, it is not unreasonable that we observed no bound P$_i$ at the $\beta_{320°}$ position under these conditions. However, to confirm the potential binding site of P$_i$ prior to being released at $\beta_{320°}$, we analyzed the structure of TF$_1$(βE190D) prepared in 100 mM phosphate buffer without the addition of MgATP (Supplementary Fig. 3c). The concentration of P$_i$ in this sample would be much higher than that expected in the cell, but having such a high concentration maximized the chance of seeing P$_i$ even if its binding affinity was low. The structure showed essentially the same conformation as the catalytic dwell, except that P$_i$ was bound to $\beta_{320°}$ instead of MgADP (Fig. 5 and Supplementary Figs. 5 and 6). We term the resultant structure the "P$_i$-bound dwell" and is most similar to the bMF$_1$ crystal structure in the presence of inhibitory

factor 1, AMP-PNP, and thiophosphate[40]. Close inspection of the $\beta_{320°}$ subunit (in both the catalytic- and P$_i$-bound dwell cryo-EM structures) indicated the presence of an internal tunnel that passes through the central cavity of the enzyme and opens only in the $\beta_{320°}$ state (Fig. 5 and Supplementary Fig. 6). As this tunnel has a minimum diameter of ~9 Å, it could potentially mediate the binding or release of P$_i$ when the path via the nucleotide-binding cleft was blocked by bound MgADP.

**The F$_1$-ATPase rotary catalytic mechanism.** The structures obtained in this study define the complete series of conformational states assumed by a β-subunit as it progresses through the F$_1$-ATPase catalytic cycle (Figs. 6 and 7, Supplementary Fig. 7, and Supplementary Movies 2 and 3). Starting at 0°—the binding-

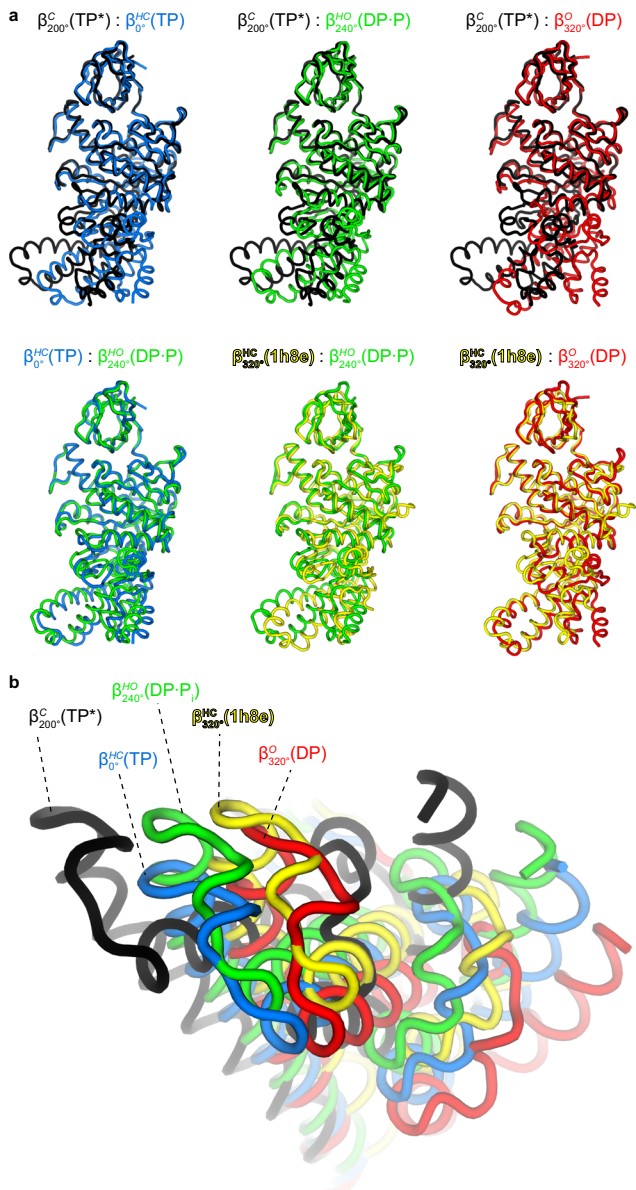

**a**
$\beta_{200°}^C(TP^*) : \beta_{0°}^{HC}(TP)$   $\beta_{200°}^C(TP^*) : \beta_{240°}^{HO}(DP\cdot P)$   $\beta_{200°}^C(TP^*) : \beta_{320°}^O(DP)$

$\beta_{0°}^{HC}(TP) : \beta_{240°}^{HO}(DP\cdot P)$   $\beta_{320°}^{HC}(1h8e) : \beta_{240°}^{HO}(DP\cdot P)$   $\beta_{320°}^{HC}(1h8e) : \beta_{320°}^O(DP)$

**b**
$\beta_{200°}^C(TP^*)$   $\beta_{240°}^{HO}(DP\cdot P_i)$   $\beta_{320°}^{HC}(1h8e)$
$\beta_{0°}^{HC}(TP)$   $\beta_{320°}^O(DP)$

**Fig. 4 Comparison of conformational states of β-subunits.** Superposition (on the N-terminal β-barrel) of the four main conformational states seen in the TF$_1$(βE190D) (this study; $\beta_{0°}^{HC}(TP)$, $\beta_{200°}^C(TP^*)$, $\beta_{240°}^{HO}(DP\bullet P)$, and $\beta_{320°}^O(DP)$), and the half-open state observed for bMF$_1$[20] (pdb1h8e). $\beta_{200°}^C(TP^*)$ (black) is in a fully closed state, $\beta_{320°}^O(DP)$ (red) is in a fully open state, $\beta_{0°}^{HC}(TP)$ and $\beta_{240°}^{HO}(DP\bullet P)$ are intermediate structures that are either half closing or half opening, and $\beta_{320°}^{HC}(1h8e)$ is an intermediate structure of bMF$_1$ similar to the open conformation. **a** Individual unique conformations compared to one other and viewed from the side. **b** Unique conformations superimposed and viewed from below.

dwell (TS) state—$\beta_{0°}^{HC}(TP)$ is in the half-closed conformation with MgATP bound loosely. After a hydrolysis event in an adjacent site, $\beta_{0°}^{HC}(TP)$ transitions from the half-closed to the $\beta_{80°}^C(TP)$ closed form through a typical induced-fit conformational transition, the so-called "binding change"[9]. The closing of the C-terminal "foot" induces an 80° rotation of the γ-subunit—the ATP-binding stroke. The torque contribution of the binding-change process is estimated to be 21–54 pN nm[41] and so is the major force-generating step. Throughout the rotation from 80° to 200°, the β-subunit remains in the closed form and acts as a solid

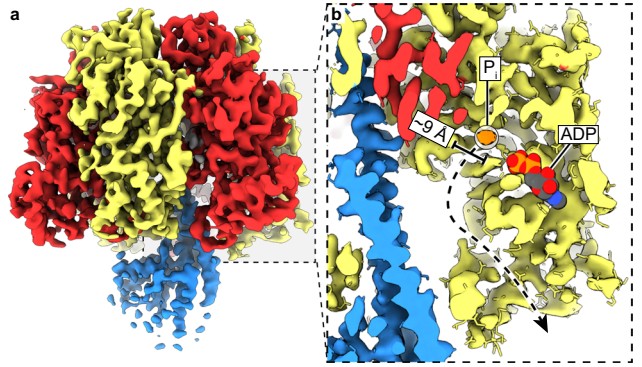

**Fig. 5 The P$_i$-bound dwell and alternative P$_i$ path. a** Cryo-EM map of the P$_i$-bound dwell. **b** Zoomed in section of the P$_i$-bound dwell with the ADP molecule from the catalytic-dwell $\beta_{320°}$ site docked and shown as spheres with CPK coloring. The alternative P$_i$ exit channel (black dashed arrow with a minimum diameter of 9 Å) could facilitate dissociation of the P$_i$ (corresponding map in orange and circled), while MgADP remains bound (also see Supplementary Fig. 6).

pivot about which the γ-subunit can rotate in response to movements in adjacent subunits. At 200°, the catalytically critical arginine residue of the α-subunit (αArg364 in TF$_1$), commonly termed the "arginine finger," is in close contact with the γ-phosphate of ATP. The β-subunit then executes a second conformational transition — from the closed form to a half-open form — resulting in the $\beta_{240°}^{HO}(DP\bullet P)$ state, with the post-hydrolysis state (ADP + P$_i$) being stabilized[42]. This conformational change results in a shift in the equilibrium to favor hydrolysis and thereby contributes to torque, albeit to a lesser extent than the binding-change process providing only 7–17 pN nm[41]. Subsequently, during the 240° to 320° rotatory movement, the β-subunit transforms a third time to a fully open conformation, the $\beta_{320°}^O$ state. After releasing ADP and P$_i$, the β-subunit then returns to the starting position at 0° where ATP binds weakly to a half-closed conformation, $\beta_{0°}^{HC}(TP)$, and the rotary cycle starts again.

The two principal movements are the ATP-binding stroke (after the binding-dwell (TS)), which is driven by the tight binding of MgATP, and the hydrolysis stroke (following the catalytic dwell), with the associated conformational changes being transferred to the γ-subunit to induce its rotation (Fig. 7). An interesting feature observed in this study is that the closing of the $\beta_{320°}^O$ subunit to the $\beta_{0°}^{HC}(TP)$ state reorients the β-subunit around the γ-subunit, so that it in a mechanically favorable position to perform the ATP-binding stroke as it closes to the $\beta_{80°}^C(TP)$ state (Fig. 7 and Supplementary Movies 2 and 3). The two trajectories that the β-subunit follows during closing can be observed the rotation axis of the "foot" of the β-subunit (residues 129–180 and 327–470) (Supplementary Fig. 8). The difference in position of the rotation axes results in the foot either pushing towards or away from the γ-subunit (open → half-closed) or twisting around the γ-subunit (half-closed → closed), with angle between these axes being ~60°. This difference is also observed in the trajectory of the γ-subunit, which wobbles (or precesses) between the two strokes rotation, observed as a change in the radius of the γ-subunit rotation between the ATP binding and hydrolysis events (Supplementary Fig. 9), which has also been seen in previous single-molecule studies[43]. A final key aspect of the F$_1$-ATPase rotary scheme is that a β-subunit remains closed and tightly bound to MgATP from 80° to 200°; this subunit acts as a solid pivot for the γ-subunit to rotate about during catalysis (Fig. 7).

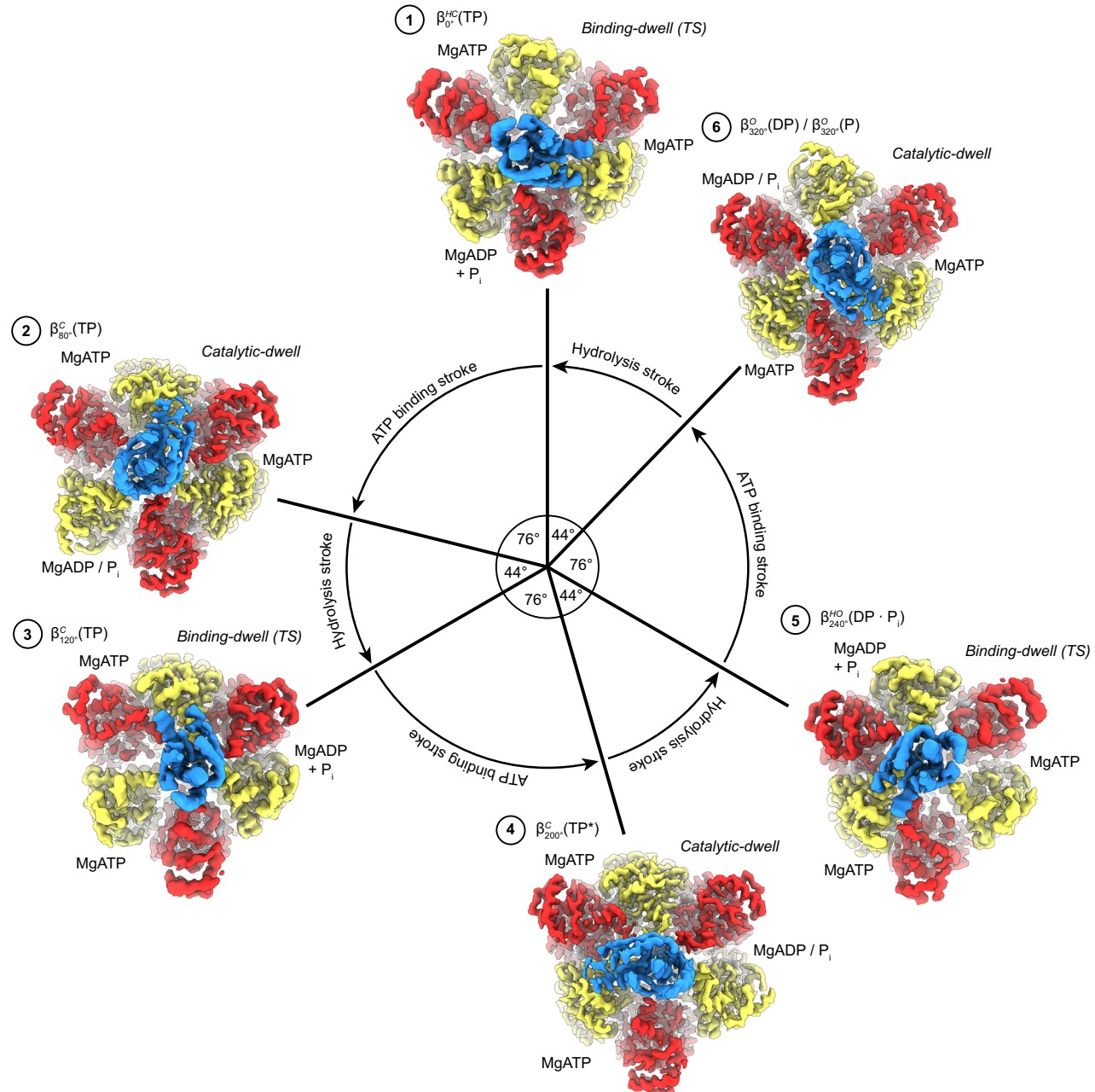

**Fig. 6 The reaction scheme of *Bacillus* PS3 F₁-ATPase suggested by cryo-EM.** Similar to Fig. 1b, the circle with arrows describes rotation of the γ-subunit. Cryo-EM maps for each state are shown as surfaces, colored as in Fig. 1a. ATP is sequentially bound and hydrolyzed, causing the ATP-binding stroke (76° rotation) and hydrolysis stroke (44° rotation). Nucleotide-binding sites are labeled 1 → 6 to highlight the reaction path of a single site.

## Discussion

Using cryo-EM, it has been possible to observe *Bacillus* PS3 F₁-ATPase in two rotational positions and correlate structural changes of the enzyme with previous single-molecule rotation experiments. Cryo-EM has the advantage of avoiding the influence of crystal lattice contacts[1,44–46] and also allowed the manipulation of temperature and buffer conditions immediately prior to freezing of the sample to reveal the binding-dwell (TS) state of the enzyme. The study also highlighted a previously unidentified tunnel that could mediate Pᵢ release, whereas the nucleotide-binding cleft remained blocked by MgADP.

The structures obtained here provide a complete rotary catalysis model for TF₁(βE190D), showing how MgATP is first tightly bound, resulting in the ATP-binding stroke, and then hydrolyzed, resulting in the hydrolysis stroke (Fig. 7 and Supplementary Fig. 10). The residues around the adenosine ring do not change substantially between any of the states, consistent with single-molecule studies on TF₁ using a base-free triphosphate[47]. Moreover, our results also indicate that Pᵢ can be released through an alternative exit channel even when the nucleotide-binding cleft is obstructed by ADP. The alternate exit path we define here is akin to the "back door" model that has been suggested to be important for force generation in myosin[48], highlighting a potential conserved mechanism between the motors. During ATP synthesis, the ability to bind nucleotide and Pᵢ through different pathways would be beneficial, preventing the system being locked if MgADP binds first.

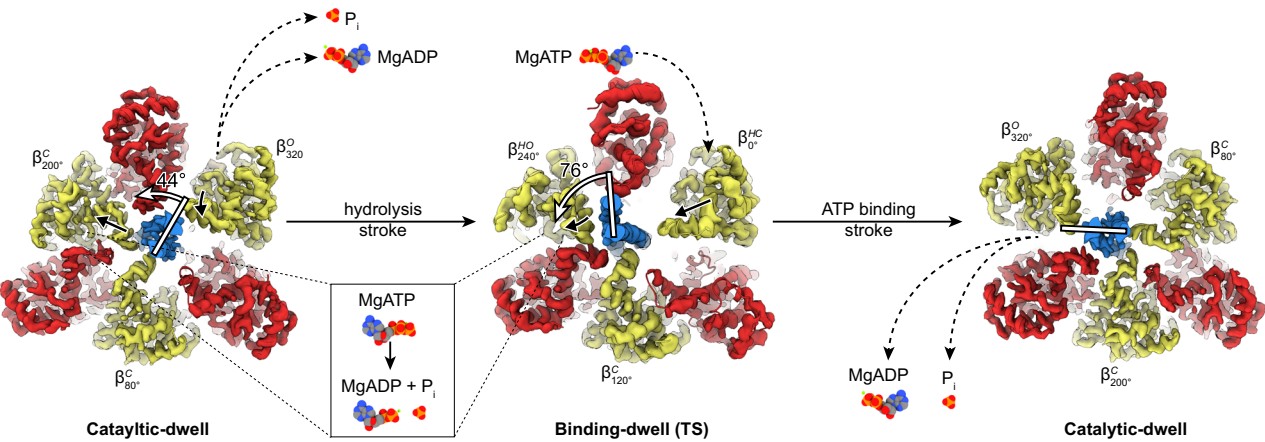

**Fig. 7 Molecular basis of F₁-ATPase substepping.** The 120° rotation of the γ-subunit is achieved by two successive steps. Black dashed arrows depict nucleotide/Pᵢ exchange in that dwell state, solid black arrows depict β-subunit movements during following stroke, and white bars with arrows depict movement of γ-subunit during following stroke. The hydrolysis stroke occurs after the catalytic dwell; MgATP is hydrolyzed to MgADP + Pᵢ in the $\beta^C_{200°}$ site, resulting in its opening to a half-open state so that the γ-subunit is pulled towards the $\beta^C_{200°}$ site and pivots around the $\beta^C_{80°}$ subunit with a 44° rotation, MgADP and Pᵢ are exchanged for MgATP in the $\beta^O_{320°}$ site, which reorients to a half-closed state primed for the ATP-binding stroke. The ATP-binding stroke occurs after the binding-dwell (TS); MgATP is tightly bound by the $\beta^{HC}_{0°}$ site, closing to a closed state, and $\beta^{HO}_{240°}$ opens to and open state, with the γ-subunit being is pushed towards the $\beta^{HO}_{240°}$ site, pivoting 76° around the $\beta^C_{80°}$(TP) subunit.

## Methods

**Protein purification.** JM103Δunc *E. coli* cells harboring the expression plasmid for PS3 F₁-ATPase (subunits α, β and δ, with a 6× His tag on β and the temperature-sensitive βE190D mutation) were incubated at 37 °C with 170 r.p.m. shaking overnight (~18 h) in a baffled flask containing 1 L Terrific broth with 100 µg/mL carbenicillin. Cells were collected at 4000 × g resulting in ~7.5 g of cells. The cell pellet was resuspended in 75 ml of 50 mM imidazole, 100 mM NaCl (pH 7.0), with one cOmplete EDTA-free Protease Inhibitor Cocktail Tablet (Roche) and DNase1 at 4 °C. Cells were lysed with sonication for 3 min at 4 °C and cell debris removed by centrifugation at 50,000 × g for 40 min at 4 °C. The supernatant was then applied to a 3 ml gravity flow Ni-NTA column that had been pre-equilibrated in 50 mM imidazole and 100 mM NaCl (pH 7.0). The column was washed with 40 column volumes of 75 mM imidazole and 100 mM NaCl (pH 7.0), and eluted with 5 column volumes of 500 mM imidazole and 100 mM NaCl (pH 7.0). Fractions containing TF₁(βE190D) (as assessed by SDS-polyacrylamide gel electrophoresis (PAGE)) were pooled and concentrated to 550 µL, before application to a Superdex 200 10/300 GL column (GE Healthcare) equilibrated in 100 mM potassium phosphate buffer and 2 mM EDTA (pH 7.0). Fractions containing TF₁(βE190D) (as assessed by SDS-PAGE) were pooled and concentrated to 8 mg/ml (300 µL) (Supplementary Fig. 1a). Standard ATP regeneration assays were performed as described in Sobti et al.[49], to ensure the protein was active (Supplementary Fig. 1b).

**Cryo-EM grid preparation.** For the phosphate buffer experiment, 3.5 µl of purified protein was transferred to a glow-discharged holey gold grid (Ultrafoils R1.2/1.3, 200 Mesh). Grids were blotted for 4 s at 22 °C, 100% humidity, and flash-frozen in liquid ethane using a FEI Vitrobot Mark IV. For the +10 mM MgATP at 28 °C or 10 °C experiments, 30 µL of protein was buffer exchanged into 20 mM Tris (pH 7.0), 50 mM KCl using a Amicon Pro spin column. Protein (4.5 µl) was then incubated at 28 °C or 10 °C in a thermocycler for 20 min. Then, 0.5 µL of 100 mM MgATP (at either 28 °C or 10 °C) was added to the protein sample and vigorously mixed with a pipette before 3.5 µl was transferred to a glow-discharged holey gold grid (Ultrafoils R1.2/1.3, 200 Mesh). Grids were blotted for 4 s at 28 °C or 10 °C, 100% humidity, and flash-frozen in liquid ethane using a FEI Vitrobot Mark IV (total time from addition of MgATP to freezing was ~20 s).

**Data collection.** Grids were transferred to a Thermo Fisher Scientific Talos Arctica transmission electron microscope (TEM) operating at 200 kV and screened for ice thickness and particle density. Grids were subsequently transferred to a Thermo Fisher Scientific Titan Krios TEM operating at 300 kV equipped with a Gatan BioQuantum energy filter (with 40 eV slit) and K2 Camera. Due to the orientation bias observed on an initial test sample, movie micrographs were recorded with tilt angles of 20–30°, as cryoEF[50] suggested an optimal angle of ~30°. Automatic data collection was performed with EPU (E Pluribus Unum - Thermo Fisher Scientific) and ×60,000 magnification (microscope user interface listed magnification of ×165,000 due to the energy filter) yielding a pixel size of 0.84 Å. A total dose of 50 electrons per Å² was used and spread over 40 frames, with a total exposure time of 5.0 s. 3156, 2965, and 2158 movie micrographs were collected for the 28 °C, 10 °C, and phosphate buffer data sets, respectively (Supplementary Fig. 3).

**Data processing.** cryoSPARC[37] was used to perform all image processing and refinement. Micrographs were first motion corrected and defocus was estimated using patches. Particles were automatically picked and subjected to two-dimensional classification to remove "junk" particles (such as particles within aggregates and minor contaminates). Trails were performed on the data to test the best way to perform heterogenous refinement by modifying the requested number of ab initio maps (between one and three), with the 28 °C data set showing three distinct structures (one of which was "junk") and the 10 °C and phosphate buffer data set showing two distinct structures. These ab initio maps were used as inputs to heterogenous refinement, which further classified the particles (see Supplementary Fig. 3). These classes were then independently refined using homogenous refinement producing the final maps. Due to some regions showing lower resolution features (particularly in the exposed region of subunit γ), DeepEMhancer[51] was used to sharpen the maps so that were more easily interpretable in the figures showing the entire complex. Supplementary Table 1 shows data collection and refinement statistics, Supplementary Fig. 11 contains FSC curves, and Supplementary Fig. 12 provides local resolution estimates. The particle numbers are summarized in Supplementary Fig. 3 and are listed here for completeness: 28 °C data set, 495,258 picked particles, 340,916 catalytic-dwell particles, 128,414 binding-dwell (TS) particles, and 25,928 "junk" particles; 10 °C data set, 727,928 picked particles, 245,378 catalytic-dwell particles, and 482,550 binding-dwell (TS) particles. Phosphate buffer data set, 479,032 picked particles, 367,412 Pᵢ-bound particles, and 111,620 α3β3 particles.

**Model building.** Models were built and refined in Coot[52], PHENIX[53], and ISOLDE[54] using pdbs 6N2Y[24] (*Bacillus* PS3 F₁Fₒ cryo-EM structure) and 4XD7[23] (*Bacillus* PS3 F₁-ATPase crystal structure) as guides. Supplementary Table 1 for refinement and validation statistics.

**Reporting summary.** Further information on research design is available in the Nature Research Reporting Summary linked to this article.

## Data availability

The models generated and analyzed during the current study are available from the protein data bank with accession codes: 7L1Q, 7L1R, and 7L1S. The cryo-EM maps used to generate models are available from the EMDB: 23115, 23116, and 23117 (DeepEMhancer[51] sharpened maps), and 24138, 24139, and 24140 (cryoSPARC[37] sharpened maps).

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

## Acknowledgements

We thank and acknowledge Dr Simon Brown for aiding in data collection and the use of the University of Wollongong Cryogenic Electron Microscopy Facility at Molecular Horizons under the management of Dr. James C. Bouwer and Directorship of Dr. Antoine van Oijen. We also wish to thank and acknowledge Dr Sawako Enoki for providing data of kinetics of TF₁(βE190D), Mr. Yi Zeng for cryo-EM data processing advice, as well as Dr Emily Furlong and Dr James Walshe for critical reading of the manuscript. We also thank and acknowledge the use of the Victor Chang Innovation Centre, funded by the NSW Government, and the Electron Microscope Unit at UNSW Sydney, funded in part by the NSW Government. Molecular graphics and analyses performed with UCSF ChimeraX, developed by the Resource for Biocomputing, Visualization, and Informatics at the University of California, San Francisco, with support from National Institutes of Health R01-GM129325 and the Office of Cyber Infrastructure and Computational Biology, National Institute of Allergy and Infectious Diseases. A.G.S. was supported by a National Health and Medical Research Council Fellowship APP1159347 and Grant APP1146403. This work was supported in part by

Grant-in-Aid for Scientific Research on Innovation Areas (JP18H04817, JP19H05380) from the Japan Society for the Promotion of Science.

## Author contributions

M.S. performed the formal analysis of the study. H.U. revised the manuscript and provided the clones and purification strategy. H.N. and A.G.S. conceived and supervised the study and drafted the manuscript.

## Competing interests

The authors declare no competing interests.
