## [Peer Review File · Nature Communications]

REVIEWER COMMENTS

Reviewer #1 (Remarks to the Author):

Sobti and colleagues present the results of a cryo-EM single particle analysis on the structure of the F1 (β E190D) complex from the thermophilic bacteria *Bacillus PS3* under three different biochemical conditions that differ in temperature (10 vs 28 degree Celsius) or content of nucleotides and phosphate (10 mM MgATP vs 100 mM Pi). Their study yielded six density maps (5 α 3 β 3 γ & 1 α 3 β 3) at a resolution sufficient to build atomic models of protein at the side chain level and the observation of larger ligands such as nucleotides or phosphate ions. They took advantage of the extensive knowledge available on the PS3 F1 complex and its β E190D mutant from both structural studies and importantly single molecule rotation assays that have established the preference for pausing at the 'catalytic dwell' at 28 degree Celsius and for the 'binding dwell' at 10 degree Celsius. This known preferential pausing during rotational catalysis allowed Sobti and colleagues to assign the major conformational state of each image data set to be either 'catalytic dwell' or 'binding dwell'. The structure of the 'binding dwell' state is described for the first time, a complete set of all six major conformations of the catalytic $\alpha\beta$ pair is now available and a new putative Pi escape channel could be identified.

This excellent study makes good use of the advantages in single particle cryo-EM to be not restricted to buffer conditions that favor 3D crystallization and to allow large scale conformational motions which are not restricted by crystal contacts — effectively providing structures for the interpretation of single molecule experimental results. This happy marriage of single particle with single molecule approaches to study rotational catalysis is an important advance which hopefully is only the first of many follow-up studies by many labs in the fields.

Before publication, however, I would like the authors to address the following points.

The very basis of this study is the separation of the mixture of conformational states that is present among the population of F1 complexes during catalytic turnover by single particle image analysis, i.e. 3D classification. This is not sufficiently well documented in the main text, the method section and the extended information. What are the respective particle numbers? How many possible states were given as input? Did the authors try several strategies? Please expand to make this crucial point of the study more transparent.

The results of the 3D classifications in terms of occupational tendency is in agreement with that of previous single molecule studies, i.e. more F1 complexes in binding dwell at 10 degree Celsius and more F1 complexes in catalytic dwell at 28 degree Celsius. The ratio, however, is differing. Please discuss critically what could be the source of that discrepancy. Did the temperature control during cryo-grid preparation not work as expected? Could there be cooling effects from rapid evaporation? Could the F1 complexes catalytic turnover be influenced by air-water interface effects?

The authors of this study are pioneers and highly respected leaders in the field of F-ATPase rotational catalysis research. While this is of benefit for the accurate interpretation of the structural findings, I feel that the introduction is not very readable to the non-expert reader who is not familiar with all the terms used in the field. To make it more accessible to a general readership please improve the writing of the introduction.

Minor points:

Please use consistent naming of the F1 complex used in this study after precise definition in the introduction. Otherwise the frequent changing of the name for the same protein complex causes confusion.

The same is true for the binding-dwell state and the catalytic-dwell state.

Please indicate for all single molecule and structural studies referred to the species, e.g. E. coli, bovine, etc., and for structures, if they were determined by X-ray crystallography or single particle cryo-EM.

As of why the TF1 mutant employed here enabled to obtain the missing binding-dwell state structure should be stated more clearly.

Practical relevant time frames for grid preparation should be stated, i.e. how long until MgADP inhibition after start of the reaction by MgATP addition; how long from MgATP addition till grid application; how long from to blotting to plunge freezing?

Figure 2 is crucial and can be improved. In all figures the yellow color used is both hard to discern and painful to the eye. Using a different yellow, using cartoon outline and/or changing the lightening setting can easily improve all figures using yellow. The six structures of Figure 2 are shown in sequential order, but their belonging to binding-dwell and catalytic-dwell is not clearly indicated. Adding arrow heads and using separate colors for the top section lines that are used to indicate the structures category can improve this problem. The stenciled parts are easy to discern, but again the in yellow depicted subunit is too hard to see.

The sentence from line 174 to 178 seems garbled.

Line 220: 'is' is missing

Figure 4: Again, please change the lightening.

Line 263: please cite 'binding change'

In Figure 5 the positions of power stroke and hydrolysis stroke are only for $\alpha\beta$ pair, but the surrounding F1 structures indicate a scheme for all three $\alpha\beta$ pairs. Please make this more clear.

Line 301-303: This sentence is garbled.

Figure 6 caption: 'two successive rotations' would mean 2×360 degrees ...

The first sentence of the discussion is a clear overstatement. Six different conformations of the $\alpha\beta$ pair is not visualizing F1 rotating by cryo-EM. Please stick to what you actually did.

Line 356: All models can be downloaded not just from the RCSB PDB.

Line 357: 7L1S is a pdb entry ID

Line 414: Why was the grid tilted?

Extended Data Fig. 2: Please make micrographs and FFT of these larger, by for example placing them on top of each other. Also indicate 3 Å resolution in the FFTs.

Extended Data Fig. 3: A more detailed caption would be an improvement.

Extended Data Fig. 4: I feel this depiction of the observed conformational changes is easier than main figure 3. What about moving it to main figure 3 as a second panel?

Extended Data Fig. 12: Please add a view from the membrane plane with a cut through at the height of the catalytic sites. Then the reader can judge more easily, if the local resolution at this decisive site

is good enough to for example distinguish between MgATP and MgADP.

Reviewer #2 (Remarks to the Author):

Sobti and coworkers present cryo-EM structures of a bacterial F1-ATPase in six different conformations. The authors exploit a temperature sensitive mutation in Bacillus PS3 F1 that slows down cooperative ATP hydrolysis to allow capture of several hitherto unseen conformations of the enzyme. Besides the X-ray crystallographically well characterized 'ground state', they were able to visualize 'binding-' and 'catalytic-dwell' states of the complex for a total of six different conformations of the catalytic beta subunits. From the structural models, the authors gain new insight into the nature of the two power strokes, the ~80deg power stroke generated when loose ATP binding is converted to tight binding, and ~40deg when ATP is hydrolyzed.

This is an exciting study that demonstrates the potential power of cryo-EM over established macromolecular crystallography in that the microscopy allows to sort and structurally characterize a distribution of conformational populations biased in a particular way by substrate and, in this case, temperature. This way, the authors managed to overcome a long standing (>20 year) bottleneck that has thus far limited our understanding of the complete catalytic cycle of the (bacterial) F1-ATPase motor from a structural standpoint (a lot is known from single-molecule observations, though). Surely, better resolution will be required to validate some of the author's interpretations regarding orientations of catalytic residues in the active site as well as nucleotide and phosphate binding (or lack thereof), as acknowledged by the authors - but overall, it seems that the authors' conclusions are justified. However, that said, the manuscript has some serious shortcomings that preclude a recommendation for publication as is. Overall, the manuscript is poorly written, both in terms of style and presentation of contents. In large part, the figures appear not crafted in a manner that will help the reader follow the dense description of the complicated catalytic mechanism with the many different conformations and catalytic site designations.

Manuscript writing (only a selection of issues is pointed out):

Lines 26-28: "F1Fo ATP synthase ... converts proton motive force (pmf) to adenosine tri-phosphate (ATP)" Surely more is needed beside pmf to make ATP.

Lines 32,33: "...in the catalytic F1-ATPase that generate ATP..."

Line 35: "...alternating ring."?

Line 35,36: "The three catalytic sites are at the interface between the alpha and beta subunits, ...". There are six interfaces - three catalytic and three non-catalytic.

Lines 100-105: Does this mean the order is ATP hydrolysis - tight binding of MgATP - Pi release? The text is difficult to follow.

Line 126,127: "...bound nucleotide in ground state structure".

Line 169: "This agrees well with 40deg substep from..."

Lines 198-201: This sentence is not clear.

Lines 211-213: Another totally unclear sentence. What does the "this" refer to? - and should the "F1-ATPase's" be F-ATPases - or not?

Line 220: "... it not unreasonable that..."

Lines 229-237: This section about the symmetric gamma less complex followed by the description of the beta320 state is unclear/confusing.

Lines 270-272: "from the closed form to a half-closed form ... resulting in generation of beta240(HO)(DP.P)" Is "half-closed" the same as "HO" - presumably half-open?

Lines 284-289: Please clarify.

Line 322: "...around the beta(TPI) subunit..." beta(TPI) is not in the figure - please define.

Figures:

- Figure 1: Please annotate the rotation of gamma with a clearly visible line or arrow etc. The figure has overall limited information content.
- Figure 2: Making the model-in-density part bigger would allow some informative annotation. Especially the density for the MgADP bound state seems not so good and assigning the tiny density above the beta phosphate is rather optimistic.
- Figure 4: The top row 1-6 (yellow-red spacefill) is hard to make out and so is the bottom row. Maybe remove unnecessary parts of the structure to allow a clearer view?
- Figure 6: The left panel shows "Hydrolysis MgATP => MgADP + Pi" but the text below says "Hydrolysis-waiting dwell". When is ATP hydrolyzed? With two sites filled - as in the left model, or with three sites filled, as in the middle model? The white arrows are hard to make out.

Other issues:

- Please define the "TS" "dwell" (line 90) or "reaction" (lines 259/60) more precisely.
- Page 12: The authors state that Pi release is from beta(320), suggesting that this state has low affinity for Pi. Does it then make sense to try to bind Pi with 100 mM phosphate? This should be clarified. Also, binding of phosphate under these conditions could be non-specific as protein associated phosphate and sulfate is often found in high-resolution crystal structures when these ions are in the crystallization cocktail.
- Please provide SDS-PAGE and ATPase activity for the preparation.
- What happened to the epsilon subunit? Should epsilon not be part of "PS3 F1-ATPase" (lines 382/3)? Or did they express only alpha3beta3gamma?

We are grateful for the reviewers' most encouraging and positive comments and for their helpful and constructive suggestions. We have modified the manuscript along the lines suggested: the text has been simplified, the figures amended, and the information requested added. A step by step response is provided below (reviewer comments in red and our replies in blue).

Reviewer #1 (Remarks to the Author):

Sobti and colleagues present the results of a cryo-EM single particle analysis on the structure of the F1 (β E190D) complex from the thermophilic bacteria *Bacillus PS3* under three different biochemical conditions that differ in temperature (10 vs 28 degree Celsius) or content of nucleotides and phosphate (10 mM MgATP vs 100 mM Pi). Their study yielded six density maps (5 $\alpha_3\beta_3\gamma$ & 1 $\alpha_3\beta_3$) at a resolution sufficient to build atomic models of protein at the side chain level and the observation of larger ligands such as nucleotides or phosphate ions. They took advantage of the extensive knowledge available on the PS3 F1 complex and its β E190D mutant from both structural studies and importantly single molecule rotation assays that have established the preference for pausing at the 'catalytic dwell' at 28 degree Celsius and for the 'binding dwell' at 10 degree Celsius. This known preferential pausing during rotational catalysis allowed Sobti and colleagues to assign the major conformational state of each image data set to be either 'catalytic dwell' or 'binding dwell'. The structure of the 'binding dwell' state is described for the first time, a complete set of all six major conformations of the catalytic $\alpha\beta$ pair is now available and a new putative Pi escape channel could be identified.

This excellent study makes good use of the advantages in single particle cryo-EM to be not restricted to buffer conditions that favor 3D crystallization and to allow large scale conformational motions which are not restricted by crystal contacts — effectively providing structures for the interpretation of single molecule experimental results. This happy marriage of single particle with single molecule approaches to study rotational catalysis is an important advance which hopefully is only the first of many follow-up studies by many labs in the fields.

Before publication, however, I would like the authors to address the following points.

Authors' response: We are most grateful for this reviewers' positive and encouraging comments on the work and are delighted to add the additional information requested.

The very basis of this study is the separation of the mixture of conformational states that is present among the population of F1 complexes during catalytic turnover by single particle image analysis, i.e. 3D classification. This is not sufficiently well documented in the main text, the method section and the extended information. What are the respective particle numbers? How many possible states were given as input? Did the authors try several strategies? Please expand to make this crucial point of the study more transparent.

Authors' response: We have added the following text to provide the additional information requested by both reviewers regarding the experimental design and execution:

Introduction:

“Cryo-EM has an advantage relative to other methods, because the environment of the sample can be controlled prior to freezing. In the present study this was exploited by changing the sample temperature immediately prior to freezing to weight the populations towards the catalytic-dwell (28°C) or binding-dwell (TS) (10°C). Subsequently

computational methods were used to sort the particles into distinct conformational states, as the data still contained heterogeneity.”

Results:

“We hypothesized that, given the results of previous single molecule observations³², this strategy of observing TF₁(βE190D) after addition of MgATP and freezing the reaction from different temperatures, would enable observation of TF₁ in two different dwell states. SPA sorting methods, termed 3D classification or heterogenous refinement, are able to sort “picked particles” to define different 3D structures or conformations within the sample³⁵. In the present study, these methods were used to separate the particles in each data set into two conformations, thereby enabling generation of cryo-EM maps of the motor in different rotational dwell states (Extended Data Fig. 3a and b).”

Methods:

“Data processing: cryoSPARC³⁷ was used to perform all image processing and refinement. Micrographs were first motion corrected and defocus was estimated using patches. Particles were automatically picked and subjected to 2D classification to remove “junk” particles (such as particles within aggregates and minor contaminants). Trials were performed on the data to test the best way to perform heterogenous refinement by modifying the requested number of *Ab initio* maps (between one and three), with the 28°C dataset showing three distinct structures (one of which was “junk”) and the 10°C and phosphate buffer dataset showing two distinct structures. These *Ab initio* maps were used as inputs to heterogenous refinement, which further classified the particles (see Extended Data Fig. 3). These classes were then independently refined using homogenous refinement producing the final maps.”

and

“The particle numbers are summarized in Extended Data Fig. 3 and are listed here for completeness: 28°C dataset, 495,258 picked particles, 340,916 catalytic-dwell particles, 128,414 binding-dwell (TS) particles and 25,928 “junk” particles. 10°C dataset, 727,928 picked particles, 245,378 catalytic-dwell particles and 482,550 binding-dwell (TS) particles. Phosphate buffer dataset, 479,032 picked particles, 367,412 Pi-bound particles and 111,620 α3β3 particles.”

The results of the 3D classifications in terms of occupational tendency is in agreement with that of previous single molecule studies, i.e. more F1 complexes in binding dwell at 10 degree Celsius and more F1 complexes in catalytic dwell at 28 degree Celsius. The ratio, however, is differing. Please discuss critically what could be the source of that discrepancy. Did the temperature control during cryo-grid preparation not work as expected? Could there be cooling effects from rapid evaporation? Could the F1 complexes catalytic turnover be influenced by air-water interface effects?

Authors’ response: We have now expanded this section to address this reviewer’s comments:

“The discrepancy between the relative proportions of the catalytic-dwell and binding-dwell (TS) states seen in this study vs. those in single molecule rotation studies, could be due to many factors. For example; (i) The classification of particles may be incomplete, with some particles assigned to the incorrect class or “junk” particles not corresponding to TF₁ molecules still being present. Much care was taken to classify these data to completion, but as particles are weighted they may still be present and not contribute to the final reconstruction, (ii) There could be cooling effects from rapid evaporation during the blotting process that would change the local temperature prior to freezing, (iii) As the protein molecules are likely

to be in contact with the air-water interface, this external influence may affect the turnover of the enzyme.”

The authors of this study are pioneers and highly respected leaders in the field of F-ATPase rotational catalysis research. While this is of benefit for the accurate interpretation of the structural findings, I feel that the introduction is not very readable to the non-expert reader who is not familiar with all the terms used in the field. To make it more accessible to a general readership please improve the writing of the introduction.

Authors’ response: We have moved Extended Data Figure 1 to the main text (now Figure 1b), added a schematic overview of F₁F₀ ATP synthase (Figure 1a). We have also changed the style of the introduction to provide a more accessible background for the non-specialist reader.

Minor points:

Please use consistent naming of the F1 complex used in this study after precise definition in the introduction. Otherwise the frequent changing of the name for the same protein complex causes confusion.

Authors’ response: We have edited the manuscript accordingly. The terms TF₁, bMF₁ and EF₁ are defined the first time they are used. We have to change between these terms, as its important to know which complexes are being discussed in different sections.

The same is true for the binding-dwell state and the catalytic-dwell state.

Authors’ response: To make the manuscript more accessible to a non-specialist readership we have removed the term “hydrolysis-waiting” dwell and changed “TS dwell” to “binding-dwell (TS)”. These are now defined more clearly in the text.

Please indicate for all single molecule and structural studies referred to the species, e.g. E. coli, bovine, etc., and for structures, if they were determined by X-ray crystallography or single particle cryo-EM.

Authors’ response: We have added this information to the text.

As of why the TF1 mutant employed here enabled to obtain the missing binding-dwell state structure should be stated more clearly.

Authors’ response: We have modified the text and added the following:

“A mutation in the catalytic-site, βE190D in TF₁, has been shown to substantially extend the duration of the binding-dwell (TS)³². Hence, as structural studies only provide a static snapshot of a given molecule at any one time, imaging TF₁(βE190D) while it undergoes ATP hydrolysis at 10°C would facilitate observation of the enzyme stalled in the binding-dwell (TS) state. Moreover, imaging the same reaction but at 28°C would produce a structure of the enzyme but in the catalytic-dwell, which could be used as a direct comparison to understand the molecular basis of rotation in the F₁-ATPase.”

Practical relevant time frames for grid preparation should be stated, i.e. how long until MgADP inhibition after start of the reaction by MgATP addition; how long from MgATP addition till grid application; how long from to blotting to plunge freezing?

Authors’ response: We have modified the results to include timing:

“There was less than 20 seconds between addition of MgATP and freezing; with 3 seconds to add/mix MgATP, 6 seconds to apply the sample to the grid and 9 seconds to blot and freeze the grid.”

Figure 2 is crucial and can be improved. In all figures the yellow color used is both hard to discern and painful to the eye. Using a different yellow, using cartoon outline and/or changing the lightening setting can easily improve all figures using yellow. The six structures of Figure 2 are shown in sequential order, but their belonging to binding-dwell and catalytic-dwell is not clearly indicated. Adding arrow heads and using separate colors for the top section lines that are used to indicate the structures category can improve this problem. The stenciled parts are easy to discern, but again the in yellow depicted subunit is too hard to see.
Authors’ response: (now Figure 3) Yellow was used as this is the traditional color used in the seminal work on F₁-ATPase. We have kept the yellow color but tried to improve the coloring/lighting/outline in the figures to increase clarity. The structures derived from either the catalytic-dwell or binding-dwell (TS) are now clearly highlighted.

The sentence from line 174 to 178 seems garbled.

Authors’ response: We have removed this sentence and combined the text to describe the new subunit notation.

Line 220: 'is' is missing

Authors’ response: Text has been updated

Figure 4: Again, please change the lightening.

Authors’ response: (now Figure 5) We have improved the figure and used the map.

Line 263: please cite 'binding change'

Authors’ response: We have cited: Boyer, P. D. The binding change mechanism for ATP synthase--some probabilities and possibilities. *Biochim Biophys Acta* **1140**, 215-250 (1993)

In Figure 5 the positions of power stroke and hydrolysis stroke are only for $\alpha\beta$ pair, but the surrounding F1 structures indicate a scheme for all three $\alpha\beta$ pairs. Please make this more clear.

Authors’ response: We have modified the figure to label all ATP binding/hydrolysis strokes

Line 301-303: This sentence is garbled.

Authors’ response: We have simplified this text and combined it with the previous sentence.

Figure 6 caption: 'two successive rotations' would mean 2x360 degrees ...

Authors’ response: The term “rotations” was not intended to mean revolutions here; hence we have changed the text to “steps”

The first sentence of the discussion is a clear overstatement. Six different conformations of the $\alpha\beta$ pair is not visualizing F1 rotating by cryo-EM. Please stick to what you actually did.

Authors’ response: The first sentence now reads: “Using cryo-EM it has been possible to observe *Bacillus* PS3 F₁-ATPase in two rotational positions and correlate structural changes of the enzyme with previous single molecule rotation experiments.”

Line 356: All models can be downloaded not just from the RCSB PDB.

Authors' response: We have changed the text to “protein data bank”

Line 357: 7L1S is a pdb entry ID

Authors' response: We have now cited the EMDB code, which was 23117.

Line 414: Why was the grid tilted?

Authors' response: The grid was tilted because our preliminary dataset showed orientation bias. We have updated the methods to read: “Due to the orientation bias observed on an initial test sample, movie micrographs were recorded with tilt angles of 20-30° as cryoEF⁴⁹ suggested an optimal angle of ~30°.”

Extended Data Fig. 2: Please make micrographs and FFT of these larger, by for example placing them on top of each other. Also indicate 3 Å resolution in the FFTs.

Authors' response: We have enlarged micrographs and power spectrums by rearranging the panels as suggested.

Extended Data Fig. 3: A more detailed caption would be an improvement.

Authors' response: We have expanded the figure and legend. It now reads; “**Comparison of the catalytic-dwell conformation with the bovine mitochondrial F₁-ATPase ground state.** (a) bMF₁ ground state crystal structure (pdb2JDI¹⁸); subunits α in dark red, β in “sand” and γ in dark blue. (b) TF₁ catalytic-dwell cryo-EM structure (from this study); subunits α in light red, β in light yellow and γ in light blue. (c) Superposition of bMF₁ ground state crystal structure (darker colors) with TF₁ catalytic-dwell cryo-EM structure (lighter colors). The structures show a similar overall fold (RMSD of 1.6 Å) in a similar rotational dwell/state.”

Extended Data Fig. 4: I feel this depiction of the observed conformational changes is easier than main figure 3. What about moving it to main figure 3 as a second panel?

Authors' response: We have moved Extended Data Fig. 4 to panel Figure 4b

Extended Data Fig. 12: Please add a view from the membrane plane with a cut through at the height of the catalytic sites. Then the reader can judge more easily, if the local resolution at this decisive site is good enough to for example distinguish between MgATP and MgADP.

Authors' response: (now Extended Data Fig. 11) We have added a view in the membrane plane to describe the local resolution in the nucleotide binding sites.

Reviewer #2 (Remarks to the Author):

Sobti and coworkers present cryo-EM structures of a bacterial F₁-ATPase in six different conformations. The authors exploit a temperature sensitive mutation in Bacillus PS3 F₁ that slows down cooperative ATP hydrolysis to allow capture of several hitherto unseen conformations of the enzyme. Besides the X-ray crystallographically well characterized ‘ground state’, they were able to visualize ‘binding-’ and ‘catalytic-dwell’ states of the complex for a total of six different conformations of the catalytic beta subunits. From the structural models, the authors gain new insight into the nature of the two power strokes, the ~80deg power stroke generated when loose ATP binding is converted to tight binding, and

~40deg when ATP is hydrolyzed.

This is an exciting study that demonstrates the potential power of cryo-EM over established macromolecular crystallography in that the microscopy allows to sort and structurally characterize a distribution of conformational populations biased in a particular way by substrate and, in this case, temperature. This way, the authors managed to overcome a long standing (>20 year) bottleneck that has thus far limited our understanding of the complete catalytic cycle of the (bacterial) F₁-ATPase motor from a structural standpoint (a lot is known from single-molecule observations, though). Surely, better resolution will be required to validate some of the author's interpretations regarding orientations of catalytic residues in the active site as well as nucleotide and phosphate binding (or lack thereof), as acknowledged by the authors - but overall, it seems that the authors' conclusions are justified. However, that said, the manuscript has some serious shortcomings that preclude a recommendation for publication as is. Overall, the manuscript is poorly written, both in terms of style and presentation of contents. In large part, the figures appear not crafted in a manner that will help the reader follow the dense description of the complicated catalytic mechanism with the many different conformations and catalytic site designations.

Authors' response: We thank the reviewer for their positive and constructive comments and are delighted that they found the study exciting. As with reviewer #1 comments, we have rewritten the text to simplify some of the concepts and remove some of the jargon. We thank the reviewer for taking the time to identify confusing points and we have changed the manuscript to rectify these shortcomings. Because the manuscript does contain a dense description of a complicated catalytic mechanism, we have now included some of the Extended Data as main figures to aid interpretation.

Manuscript writing (only a selection of issues is pointed out):

Lines 26-28: "F₁F_o ATP synthase ... converts proton motive force (pmf) to adenosine tri-phosphate (ATP)" Surely more is needed beside pmf to make ATP.

Authors' response: To clarify this, we have modified the text to read: "F₁F_o ATP synthase is a biological rotary motor that utilizes a rotary catalytic mechanism to couple proton translocation across a membrane with the synthesis of adenosine tri-phosphate (ATP) from inorganic phosphate (P_i) and adenosine di-phosphate (ADP)¹⁻⁴"

Lines 32,33: "...in the catalytic F₁-ATPase that generate ATP..."

Authors' response: We have edited the text to: "This central rotor drives conformational changes in the catalytic F₁-ATPase, where ATP is synthesized from ADP and P_i^{5,6}."

Line 35: "...alternating ring."?

Authors' response: We have edited the text to: "The F₁-ATPase consists of three α , three β and a single γ subunit, with the α and β subunits arranged in an alternating fashion ($\alpha\beta\alpha\beta\alpha\beta$) to form a hexameric ring with the γ subunit at its center (Extended Data Fig. 1)". With this new Extended Data Figure used to introduce the general architecture of ATP synthase.

Line 35,36: "The three catalytic sites are at the interface between the alpha and beta subunits, ...". There are six interfaces - three catalytic and three non-catalytic.

Authors' response: We have edited the text to: "The F₁-ATPase consists of three α , three β and a single γ subunit, with the α and β subunits arranged in an alternating fashion ($\alpha\beta\alpha\beta\alpha\beta$) to form a hexameric ring with the γ subunit at its center (Fig. 1a). There are six nucleotide binding sites at the interfaces between the α and β subunits, with three being catalytically

active and three being non-catalytically active. The three catalytically active sites are mainly encompassed by the β subunits and therefore are termed the β subunit sites.”

Lines 100-105: Does this mean the order is ATP hydrolysis - tight binding of MgATP - Pi release? The text is difficult to follow.

Authors’ response: We have edited the text to: “The cryo-EM maps presented in this study indicate how ATP tightly binds to induce the 80° ATP binding stroke and ATP hydrolysis induces the 40° hydrolysis stroke.”

Line 126,127: “...bound nucleotide in ground state structure”.

Authors’ response: We have edited the text to: “Although the structure was highly similar to the crystal structure of bMF₁ in the ground state, one clear difference was the β_E nucleotide occupancy. In the crystal structure of bMF₁ in ground state, the β_E site was empty and did not contain any nucleotide, whereas the equivalent site in the cryo-EM structure of TF₁ contained MgADP.”

Line 169: “This agrees well with 40deg substep from...”

Authors’ response: We have edited the text to: “This agrees well with the 40° rotation observed between the catalytic-dwell and binding-dwell (TS) in single molecule studies^{15,32}, suggesting that the predominate conformation observed when frozen from 10°C represents the state pausing at the binding-dwell angle, as expected.”

Lines 198-201: This sentence is not clear.

Authors’ response: We have addressed this text in response to reviewer #1 and expanded on this idea. This section now reads: “The discrepancy between the relative proportions of the catalytic-dwell and binding-dwell (TS) states seen in this study vs. those in single molecule rotation studies, could be due to many factors. For example; (i) The classification of particles may be incomplete, with some particles assigned to the incorrect class or “junk” particles not corresponding to TF₁ molecules still being present. Much care was taken to classify these data to completion, but as particles are weighted they may still be present and not contribute to the final reconstruction, (ii) There could be cooling effects from rapid evaporation during the blotting process that would change the local temperature prior to freezing, (iii) As the protein molecules are likely to be in contact with the air-water interface, this external influence may affect the turnover of the enzyme.”

Lines 211-213: Another totally unclear sentence. What does the “this” refer to? - and should the “F1-ATPase’s” be F-ATPases - or not?

Authors’ response: We have edited the text to: “The set of sequential conformational states observed in this study are broadly consistent with the scheme suggested by single molecule experiments that is outlined in Fig. 1b. However, an exception is that ADP is observed bound to the β_{320° site, whereas previous studies, based on information from the ground state bMF₁ crystal structure, hypothesized that this site would be empty and not contain any nucleotide. One potential interpretation of this point is that this discrepancy represents a divergence of the F₁-ATPase reaction scheme between species and the β_{320° conformational state represents a novel catalytic intermediate, an “ADP-releasing state” that is only present in TF₁.”

Line 220: “... it not unreasonable that...”

Authors’ response: We have edited the text to: “...it is not unreasonable that...”

Lines 229-237: This section about the symmetric gamma less complex followed by the description of the beta320 state is unclear/confusing.

Authors' response: We have removed the section on the symmetric gamma less complex to aid clarity.

Lines 270-272: “from the closed form to a half-closed form ... resulting in generation of beta240(HO)(DP.P)” Is “half-closed” the same as “HO” - presumably half-open?

Authors' response: We have now defined the terms HC and HO when we introduce the naming method. The conformations are not the same. The half-closed and half-open conformations differ as shown in our updated Figure 3 (now Figure 4) and Movie 3.

Lines 284-289: Please clarify.

Authors' response: We have removed this text for clarity and now do not mention the open state seen in the symmetric structure.

Line 322: “...around the beta(TPI) subunit...” beta(TPI) is not in the figure - please define.

Authors' response: This was a typo that was carried forward from a previous manuscript draft. Beta(TPI) has been replaced with $\beta_{80^\circ}^C(TP)$

Figures:

- Figure 1: Please annotate the rotation of gamma with a clearly visible line or arrow etc. The figure has overall limited information content.

Authors' response: (now Figure 2) We have increased the size of the arrow and added a white bar to show rotation. We would like to include this figure to show that both the catalytic-dwell and binding-dwell (TS) structures have been solved.

- Figure 2: Making the model-in-density part bigger would allow some informative annotation. Especially the density for the MgADP bound state seems not so good and assigning the tiny density above the beta phosphate is rather optimistic.

Authors' response: (now Figure 3) We did include Movie 1 in the original submission which provides rocking views of the individual nucleotide sites. The figure is already a full-page figure, but we have zoomed in on the sites and included larger versions of the original panels in the supplementary (Extended Data Figure 4) to aid readership.

- Figure 4: The top row 1-6 (yellow-red spacefill) is hard to make out and so is the bottom row. Maybe remove unnecessary parts of the structure to allow a clearer view?

Authors' response: We have removed these panels and they were no longer deemed necessary.

- Figure 6: The left panel shows “Hydrolysis MgATP => MgADP + Pi” but the text below says “Hydrolysis-waiting dwell”. When is ATP hydrolyzed? With two sites filled - as in the left model, or with three sites filled, as in the middle model? The white arrows are hard to make out.

Authors' response: We have edited this figure to improve clarity.

Other issues:

- Please define the “TS” “dwell” (line 90) or “reaction” (lines 259/60) more precisely.

Authors' response: In response to this and the other comments from reviewer #1 we have tried to simplify the naming of these dwells and introduced them more clearly. We have also changed the term “TS dwell” to “binding-dwell (TS)”.

- Page 12: The authors state that Pi release is from beta(320), suggesting that this state has low affinity for Pi. Does it then make sense to try to bind Pi with 100 mM phosphate? This should be clarified. Also, binding of phosphate under these conditions could be non-specific as protein associated phosphate and sulfate is often found in high-resolution crystal structures when these ions are in the crystallization cocktail.

Authors' response: This experiment was designed to see the potentially binding site of Pi in this rotatory state. We could have just inferred it from the known gamma phosphate binding residues, or by using the crystal structure of bMF1 incubated with thiophosphate, but we felt it was more meaningful to solve the cryo-EM structure under an extremely high concentration of Pi. Furthermore, TF1 has been shown to be highly stable in 100 mM phosphate, given the complex is purified in this buffer (see the methods) and therefore it seemed like the ideal conditions to perform this experiment. We have edited the manuscript to: “The concentration of Pi in this sample would be much higher than that expected in the cell, but having such a high concentration maximized the chance of seeing Pi even if its binding affinity was low.”

- Please provide SDS-PAGE and ATPase activity for the preparation.

Authors' response: We have added a new figure (Extended Data Figure 2) to show an SDS-PAGE and ATP regeneration assays.

- What happened to the epsilon subunit? Should epsilon not be part of “PS3 F1-ATPase” (lines 382/3)? Or did they express only alpha3beta3gamma?

Authors' response: We only expressed the alpha3beta3gamma complex. This is now described in the results and methods, with an explanation of why the epsilon subunit was not included has been added to the introduction: “In studies examining the isolated bacterial F1-ATPase undergoing hydrolysis, such as the study presented here, the ϵ subunit is usually not expressed and purified so that it does not inhibit the enzyme⁷.”

REVIEWER COMMENTS

Reviewer #1 (Remarks to the Author):

I am happy to see that my own and the concerns of Reviewer #2 are now carefully addressed. The context and the findings of this single particle cryoEM study on the catalytic cycle of F1 ATPase are described and presented in a much more clear fashion that is suitable for the wider readership of Nature Communications.

I have now little to add, but would be glad to see the following minor points taken care of before publication:

- the second quotation mark in line 60 is missing
- in line 122 'ATP binding angle' might be better than just 'binding angle'
- line 134-136: this is also true for X-ray crystallography, especially XFEL experiments where caged substances are released shortly before or during data acquisition. Actually given that X-ray crystallography can easily performed at room temperature and atmospheric pressure, it is easy to argue that X-ray crystallography is superior to single particle cryo-EM in regard to its ability to change buffer, substrate, temperature conditions shortly prior or even during measurements. It is the necessity of crystal contacts that mainly restrict X-ray crystallography.
- line 153: 'catalytic dwell angle' should be better
- line 185: in line 260 'below' is defined as 'as viewed from the membrane plane for the whole F1Fo complex'; perhaps here it is more suitable place to introduce this definition of 'below'
- line 216: it should be 'at 28 degree'
- line 229: 'cryo-EM map' should be replaced by either 'density' or 'Coulomb density', since the cryo-EM map is actually the whole box including not just a nucleotide
- line 262: 'cryo-EM image data set' might be better
- line 265: 'this ratio' might be better than just 'this'

Reviewer #2 (Remarks to the Author):

This is a revised manuscript. The authors have addressed the majority of issues raised in the first round of review satisfactorily and as a result, the manuscript text and figures are much improved. However, a few of the revisions and authors' responses require further clarification as follows:

- Lines 38-40: "...three being non-catalytically active". What is non-catalytic activity?
- Line 132: "...temperate resolved..."?
- Line 172: "predominate" => predominant?
- Line 175: in an open conformation
- New figure 5: While the EM density at the contour drawn indicates a potential tunnel for Pi release, it appears that the amino acid side chains that line the tunnel are not in density. Is the tunnel still present and large enough for Pi when the tunnel lining side chains are displayed in space-fill?

We are again very grateful for the reviewers' positive and constructive comments. We have performed the small modifications suggested by both reviewers and have included a further Extended Data Figure to answer Reviewer #2's query on space-filling. A step by step response is provided below (reviewer comments in red and our replies in blue).

REVIEWER COMMENTS

Reviewer #1 (Remarks to the Author):

I am happy to see that my own and the concerns of Reviewer #2 are now carefully addressed. The context and the findings of this single particle cryoEM study on the catalytic cycle of F1 ATPase are described and presented in a much more clear fashion that is suitable for the wider readership of Nature Communications.

I have now little to add, but would be glad to see the following minor points taken care of before publication:

- the second quotation mark in line 60 is missing

We have added the second quotation mark

- in line 122 'ATP binding angle' might be better than just 'binding angle'

We have changed this to the "ATP binding angle"

- line 134-136: this is also true for X-ray crystallography, especially XFEL experiments where caged substances are released shortly before or during data acquisition. Actually given that X-ray crystallography can easily performed at room temperature and atmospheric pressure, it is easy to argue that X-ray crystallography is superior to single particle cryo-EM in regard to its ability to change buffer, substrate, temperature conditions shortly prior or even during measurements. It is the necessity of crystal contacts that mainly restrict X-ray crystallography.

We have changed the words to remove the notion that cryo-EM is advantageous over crystallography in this sense.

New text: "In cryo-EM it is relatively trivial to control the sample environment prior to freezing."

- line 153: 'catalytic dwell angle' should be better

We have changed the words improve the flow of the sentence: "...pausing at both the binding-dwell and catalytic-dwell angles..."

- line 185: in line 260 'below' is defined as 'as viewed from the membrane plane for the whole F1Fo complex'; perhaps here it is more suitable place to introduce this definition of 'below'

We now define "below" in the figure legend of Fig.1: "...viewed from the membrane, which is referred to as the "below" view hereafter..." and have removed the definition from line 260.

- line 216: it should be 'at 28 degree'

We have changed "a" to "at"

- line 229: 'cryo-EM map' should be replaced by either 'density' or 'Coulomb density', since the cryo-EM map is actually the whole box including not just a nucleotide

We have changed "cryo-EM map" to "Coulomb density"

- line 262: 'cryo-EM image data set' might be better

We have added "cryo-EM image" to the term "dataset"

- line 265: 'this ratio' might be better than just 'this'

We have changed this to "...this ratio of particles..."

Reviewer #2 (Remarks to the Author):

This is a revised manuscript. The authors have addressed the majority of issues raised in the first round of review satisfactorily and as a result, the manuscript text and figures are much improved. However, a few of the revisions and authors' responses require further clarification as follows:

- Lines 38-40: "...three being non-catalytically active". What is non-catalytic activity?

We have changed the text to state that it is the nucleotide binding sites which are not active: "There are six nucleotide binding sites at the interfaces between the α and β subunits, with three of the nucleotide binding sites being catalytically active and the other three nucleotide binding sites being non-catalytically active."

- Line 132: "...temperate resolved..."?

We have changed "temperate" to "temperature"

- Line 172: "predominate" => predominant?

We have changed "predominate" to "predominant"

- Line 175: in an open conformation

We have added the word "an"

- New figure 5: While the EM density at the contour drawn indicates a potential tunnel for Pi release, it appears that the amino acid side chains that line the tunnel are not in density. Is the tunnel still present and large enough for Pi when the tunnel lining side chains are displayed in space-fill?

The previously submitted figure showed that, when the structure is presented as a surface the tunnel is large enough for Pi to escape via the alternative exit tunnel. We have added a new Extended Data figure (Extended Data Fig. 6) to show surface representation of this view as well.